# Achieving the UNAIDS first '95' in prenuptial HIV/AIDS testing among reproductive-aged Rwandese women: A multilevel analysis of 2019–20 population-based data

**Michael Ekholuenetale**[1], **Olah Uloko Owobi**[2], **Amadou Barrow**[3]*

**1** Department of Epidemiology and Medical Statistics, Faculty of Public Health, College of Medicine, University of Ibadan, Ibadan, Nigeria, **2** Faculty of Medicine, College of Medical Sciences, University of Maiduguri, Maiduguri, Nigeria, **3** Department of Public & Environmental Health, School of Medicine & Allied Health Sciences, University of The Gambia, Kanifing, The Gambia

* abarrow@utg.edu.gm

**Data Availability Statement:** Data for this study were sourced from the 2019-20 Individual Woman Recode Questionnaire from Rwanda Demographic

## Abstract

A significant public health concern that disproportionately affects women is human immuno-deficiency virus (HIV). Prenuptial HIV testing is no doubt a major step for people to learn their HIV status. In this study, the coverage of prenuptial HIV testing and its associated factors were examined among reproductive-aged Rwandese women. This study included a total of 14,634 reproductive-aged Rwandese women using 2019–20 Rwanda Demographic and Health Survey (RDHS). The coverage of prenuptial HIV/AIDS testing and the variables influencing it were evaluated using percentage and multilevel logit model. The level of significance was set at $p<0.05$. The weighted prevalence of prenuptial HIV/AIDS testing was 45.9% (95%CI: 44.8%-47.1%). The respondents who attained primary and secondary+ education had 31% (OR = 1.31; 95%CI: 1.09–1.59) and 56% (OR = 1.56; 95%CI: 1.25–1.95) higher odds of prenuptial HIV/AIDS testing, when compared with uneducated women. Those who got married or had their first sex at an adult age (18+ years), had higher odds of prenuptial HIV/AIDS testing, when compared with women who got married before age 18 years or never had sex respectively. Women's age, nativity and region were associated with prenuptial HIV testing. Women with knowledge of HIV test kits, had higher odds of prenuptial HIV/AIDS testing (OR = 1.45; 95%CI: 1.30–1.63), when compared with those with no knowledge of HIV test kits. The respondents from female-headed households had 12% reduction in prenuptial HIV/AIDS testing (OR = 0.88; 95%CI: 0.80–0.97), when compared with their male-headed counterparts. The moderately (OR = 1.16; 95%CI: 1.03–1.31) and highly (OR = 1.55; 95%CI: 1.37–1.75) enlightened women had higher odds of prenuptial HIV/AIDS testing, when compared with those with low enlightenment. The uptake of prenuptial HIV/AIDS testing was relatively low among Rwandese women. We recommend improving women's education, enlightenment, delay in sexual debut, marriage at adult age (18years) and increasing knowledge about HIV testing among women.

and Health Survey (RDHS). The variables selected from the dataset have been presented in the - selection and measurement of variables section of the methods. Please, see here to download: https://www.dhsprogram.com/data/dataset/Rwanda_Standard-DHS_2019.cfm?flag=1.

**Funding:** The authors received no specific funding for this work.

**Competing interests:** The authors have declared that no competing interests exist.

## Introduction

Approximately 17.8 million women who are 15+ years old are estimated to have contracted human immunodeficiency virus (HIV), making up approximately 52% of all adults living with HIV globally [1]. A single year highest new cases of HIV infection in 1999 was 3.16 million, however, by 2017 it declined to 1.94 million [2]. Regional variations remain in both new HIV infections and the 37.9 million people living with HIV, with Africa accounting for 25.7 million of the total [3]. Women account for approximately 56% of all new HIV cases among adults in sub-Saharan Africa (SSA) (aged 15 years and older). Young women and adolescent girls could have higher likelihood than their male counterparts to contract HIV. Women and adolescent girls reported about 63% of all new cases of HIV infections in SSA [4]. HIV was a top cause of death for reproductive-aged women in 2017 [5]. In 2020, more than 4000 young women and adolescent girls between 15–24 years old, contracted HIV weekly [4]. The leading countries in HIV are located in SSA with prevalence rates ranging from 8.9 to 27% [6].

Despite the magnitude of HIV incidence among women, the use of HIV testing services remains suboptimal [7]. The United Nations Programme on HIV/AIDS (UNAIDS) established a global 95-95-95 strategy of reaching 95% of HIV positive people to be aware of their status, enroll 95% of HIV positive individuals in antiretroviral therapy (ART), and have viral suppression for 95% of those on ART by 2030 [8]. The prenuptial HIV/AIDS testing is essential because it ensures continued progress towards achieving the UNAIDS first '95' [8]. Since the outbreak, approximately 79.3 million people have contracted HIV by 2020, about 36.3 million have died due to diseases related to HIV/AIDS [9]. In 2020, about 1.5 million new HIV infections were reported, 680 000 deaths were linked to HIV/AIDS, 37.7 million people living with HIV, and 84% of HIV positive people knew their status [9]. The global incidence of HIV in adults between the ages of 15 and 49 years, varies greatly across different regions and countries [6]. The United Nations General Assembly high-level meeting on AIDS adopted the 2016 Political Declaration on HIV/AIDS, which reaffirmed the 2030 Agenda for Sustainable Development Goals (SDGs). Seven of the 17 SDGs, including ending the AIDS epidemic by 2030, are relevant to a global and national HIV response [10, 11].

HIV has remained endemic in many resource-constrained settings, owing to a variety of factors, such as women's large number of sexual partners [12]. According to a study conducted in Rwanda among young women and adolescent girls, education, wealth, access to media, ever had sex and HIV knowledge were linked with HIV testing [13]. In evidence-based reports, up to 50% of adolescent girls' sexual debuts was due to violence and raises the chance of contracting HIV [14–16], as women cannot negotiate for safe sex in the instance of sexual violence. Similarly, early marriage would lower women empowerment and access to HIV testing and prevention information [17]. Women have less chances to get tested for HIV, for a variety of reasons including gender inequality, inadequate knowledge of the disease, lack of access to diagnostic services, low educational attainment and extreme poverty [18]. The practice of premarital HIV testing can be influenced by urban residence, education, access to media, wealth, knowledge of the place of HIV testing and lifestyle [19]. Also, previous testing for HIV, age and working status are known factors influencing premarital HIV testing [20].

Rwanda has 3% HIV prevalence among adults [21]. In Rwanda, HIV prevalence is approximately 4% among reproductive-aged women, being 2% higher than their male counterparts [21, 22]. A recent study conducted in Rwanda, found inadequate HIV testing (approximately 55.4%) among young women and adolescent girls [13]. In another study, about 60% of young women between 15 to 24 years old, who were sexually active within the year before the survey had tested for HIV [22]. In terms of speeding up the scale-up of HIV testing, Rwanda's national HIV program has made some progress, as about 83.8% of HIV-positive adults aged

15 to 64 years have self-reported HIV testing, with 85.6% and 80.4% being women and men reporting HIV-testing respectively [23]. This indicates that the country has made significant progress in HIV testing and counseling, implying that women have higher HIV testing rates than men. The need for timely HIV testing in women still exists, and understanding of the factors that can influence HIV testing could help UNAIDS reach its first '95 goal. This study looked at the coverage of prenuptial HIV testing and the individual compositional and community-level factors associated with it among reproductive-aged Rwandese women.

## Methods

### Data source

Using a large size secondary dataset, hierarchical model analysis was carried out. For this study, data from individual woman questionnaires in the 2019–20 Rwanda Demographic and Health Survey (RDHS) were extracted. A total sample of 14,634 reproductive-aged women were examined. The 2019–20 RDHS was the sixth round following the 1992, 2000, 2005, 2010, and 2014–15 surveys. The survey was carried out by the National Institute of Statistics of Rwanda, supported by Ministry of Health and Inner City Fund (ICF). The survey was conducted November, 2019 through July, 2020. Because of the impact of lockdown following coronavirus pandemic in 2020, data collection was halted for about three months (March-June) [24]. The RDHS gathered information on factors such as fertility rates and preferences, use of contraception, maternal and child health, rates of mortality, nutrition, knowledge of HIV/AIDS, STIs amongst others relevant to monitor population health [24]. A previous study has reported the methodology of DHS [25].

### Sampling design

A 2-stage stratified cluster sampling was adopted in the 2019–20 RDHS, with the goal of allowing important indicators for the country to be estimated, while considering urban versus rural residence, geographical region (5 divisions) and districts (30 in Rwanda) for selected metrics. The National Institute of Statistics, RDHS's implementing agency, provided a sampling frame of enumeration areas (EAs) for the entire country. EA is a community established for the 2012 Rwanda Population and Housing Census used as census counting unit. In the first step, sample clusters made up of EAs designated for the 2012 Rwanda Population and Housing Census were chosen. A total of 500 clusters were chosen, with 388 in rural and 112 in urban residence. Systematic household sampling was done in the second phase. From June to August 2019, a household listing was conducted in all chosen EAs, and households included in the survey were randomly chosen. Each cluster had approximately 26 households, thereby resulting to 13,000 households across the country.

### Selection and measurement of variables

**Outcome.** The dependent variable for this study was measured in binary form and coded as "*1*" if a woman responded "*yes*" to the question: "*Ever tested for AIDs for prenuptial purposes*" and "*0*" if otherwise.

**Individual-level factors.** The variables selected for this studies came from previous studies that looked at HIV positivity, testing and associated variables in women of childbearing age in African settings [26–28]. Women's age (in years): <25/25-34/over 34; educational attainment: uneducated/primary/secondary or higher; working status: not employed/employed; age at first union or marriage (in years): <18/18-24/over 24; age at first sex (in years): never had sex/<18/18+; health insurance: yes/no; years lived in place of residence: 5+/<5; literacy:

cannot read at all/able to read; household head: female versus male; knowledge of HIV test kits: no/yes; media access: no/yes; religion: Catholic/other Christians/Muslims/others; socioeconomic status: included rural, poorest, lack of formal education and unemployment. The measure was developed by principal component analysis, with the score classified to low/ medium/high. Enlightenment: using principal component analysis with the indicators of women having formal education, exposure to print media, radio, television and internet. This was categorized as: low/medium/high. Wealth quintiles: The weights for the wealth indicators were determined using principal component analysis as reported in previous studies [29, 30].

**Community-level factors.** Due to the DHS's lack of community-level data collection, we prominently represented communities using EAs. The community-level measures were therefore on individual's characteristics, especially when it has potential impact on prenuptial HIV testing. Women's place of residence: rural versus urban. Geographical region: Kigali/South/ West/North/East.

Additionally, by combining individual level characteristics at cluster level, aggregate measures were created. The aggregate variables were then classified as low or high depending on the distribution of the proportion values calculated for each cluster. To examine the distribution of the proportion values, a normality test was performed. If the aggregate measure was normal, the mean value was used as the cut-off point for categorization; if it was skewed, the median value was used instead. When women from poorest and poorer wealth categories in a cluster made up 38–100% of the population, it was considered to have high levels of poverty, and those having 0–37% were considered to have low levels of poverty. Media use at the community level was classified as high if the percentage of women users was 86–100% and as low if it was 0–85%. Community-level illiteracy was classified as high if the percentage of women who are completely unable to read was greater than or equal to 13–100%, and as low if it was less than or equal to 0–12%. Community-level nativity was categorized as high if proportion of women who lived in the place of residence 5+ years was 76–100% and as low if the proportion of women who lived in the place of residence 5+ years was 0–75%. Community-level HIV knowledge of test kits was categorized as high if proportion of women who knew about HIV test kits was 16–100% and as low if the proportion of women who did not know about HIV test kits was 0–15%. We adopted similar approach as used in previous studies [31–33].

**Ethical consideration.** A secondary dataset available in the public domain with identifier information removed, was analyzed for this study. DHS followed a standard ethical procedure to obtain informed consent from respondent. There was no need for additional participants' consenting, as the authors obtained permission to use the dataset for the purpose of this study. The information regarding DHS ethical standards can be found here: http://goo.gl/ny8T6X.

## Statistical analysis

To compute the estimates of prenuptial HIV/AIDS testing, Stata 'svy' module was utilized in adjusting for clustering, stratification and sampling weights. The prevalence of prenuptial HIV/AIDS testing was explored using percentage. A variance inflation factor analysis was used to examine the presence of multicollinearity [34]. In the absence of multicollinearity, significant variables from the Chi-square test were retained in the logit model.

The mixed-effects of the factors associated with prenuptial HIV/AIDS testing were estimated using an adjusted multilevel binary logit model. For binary response reporting prenuptial HIV/AIDS testing, we developed a two-level model for individual women (at level 1) and communities/enumeration areas (at level 2). These were the levels with statistically significant variances. We created four models. The null or empty model without any factor, was created to examine community-level variance. This is used to understand how much community

factors could explain the observed variations. When the variance is statistically significant, the use of a multilevel model is justified. If the community-level variance was not significant, the single-level logit model is recommended. The second model contains individual compositional factors only, while the third model has the cluster-level factors only. The fourth model adjusted for individual compositional and cluster/community-level factors. The statistical significance level was set at 5%. The Bayesian or Akaike Information Criterion was used to choose the best model from among the four options. A lower Akaike or Bayesian Information Criterion value indicates a better model fit [35]. Stata Version 16 was utilized for data analyses.

### Fixed and random effects

Adjusted odds ratios (AORs) with their 95% confidence intervals were used to report the findings of fixed effects (measures of association) (CI). There is a 95% chance that the parameter will take a value within the given range, according to a 95% confidence interval [36]. The Intra-class Correlation (ICC) and Median Odds Ratio (MOR) were used to assess the likely contextual effects [37]. ICC was used to assess the similarity of respondents in the same household and community. The ICC is a measure of the clustering of odds of prenuptial HIV/AIDS testing in the same community. It represents a percentage of the total variance in the probability of prenuptial HIV/AIDS testing that is related to the community level. The MOR calculates the odds ratio for the second level variance and estimates the likelihood of prenuptial HIV/AIDS testing that can be attributed to community context. MOR of unity indicates that there is no variation in the household or community. In contrast, the higher the MOR, the more significant the contextual effects for understanding the likelihood of prenuptial HIV/AIDS testing. According to Snijders and Bosker's formula [38], the ICC was determined using the linear threshold, while the MOR is a measurement of cluster heterogeneity that is not explained.

### Results

The weighted prevalence of prenuptial HIV/AIDS testing was 45.9% (95% CI: 44.8–47.1).

Table 1 shows that women aged 25–34 years reported the prevalence of prenuptial HIV/AIDS testing of 68.7%. Educated women reported an increased prevalence of prenuptial HIV/AIDS testing. Currently employed women reported 49.4% of prenuptial HIV/AIDS testing, those not employed had 37.8% of prenuptial HIV/AIDS testing. Women who had first sex or got married at an adult age (18+ years), reported higher prevalence of prenuptial HIV/AIDS testing respectively. Those who lived in the place of residence <5 years prior to the survey, reported 56.8% of prenuptial HIV/AIDS testing. Respondents who had knowledge of HIV test kits reported 56.4% prevalence of prenuptial HIV/AIDS testing. See Table 1 below for the details.

### Measures of variations and model fit statistics

In Table 2, values from the unconditional model (Model I) showed that there was a significant variation in the odds of prenuptial HIV/AIDS testing across communities ($\sigma^2$ = 0.26; 95% CI: 0.21–0.31). Model IV (full model) was selected as the most suitable due to the least AIC value (14948.1). The variations in the odds of prenuptial HIV/AIDS testing across communities ($\sigma^2$ = 0.32; 95% CI: 0.26–0.40) was estimated. Results from Median Odds Ratio became the evidence of community contextual factors shaping prenuptial HIV/AIDS testing. It was estimated that if a women moved to another community with a higher probability of prenuptial HIV/AIDS testing, the median increase in their odds of prenuptial HIV/AIDS testing would be 1.71 with ICC of 8.8%. The full model accounted for -23.1% of the variance in the odds of prenuptial HIV/AIDS testing across the community. This implied that small amount of variance in

**Table 1. Prevalence of prenuptial HIV/AIDS testing across women's characteristics (n = 14,634).**

| Variable | Number of women (%) | Weighted prevalence of prenuptial HIV/AIDS testing; % (95%CI) | P value |
|---|---|---|---|
| **Age (in years)** | | | <0.001* |
| <25 | 5732 (39.2) | 25.9 (24.7–27.0) | |
| 25–34 | 4142 (28.3) | 68.7 (67.2–70.1) | |
| >34 | 4760 (32.5) | 48.5 (47.0–49.9) | |
| **Education** | | | <0.001* |
| No formal education | 1352 (9.2) | 41.8 (39.2–44.5) | |
| Primary | 8500 (58.1) | 47.2 (46.1–48.2) | |
| Secondary+ | 4782 (32.7) | 43.6 (42.2–45.0) | |
| **Wealth quintiles** | | | <0.001* |
| Poorest | 2844 (19.4) | 42.9 (41.1–44.8) | |
| Poorer | 2707 (18.5) | 43.1 (41.2–45.0) | |
| Middle | 2709 (18.5) | 45.8 (43.9–47.7) | |
| Richer | 2884 (19.7) | 47.8 (45.9–49.6) | |
| Richest | 3490 (23.9) | 47.3 (45.6–49.0) | |
| **Working status** | | | <0.001* |
| Not employed | 4947 (33.8) | 37.8 (36.5–39.2) | |
| Employed | 9687 (66.2) | 49.4 (48.4–50.4) | |
| **Age at first marriage (in years)** | | | <0.001* |
| <18 | 989 (6.8) | 40.1 (37.1–43.2) | |
| 18–24 | 5670 (38.8) | 61.7 (60.4–62.9) | |
| >24 | 1915 (13.1) | 79.1 (77.2–80.9) | |
| Never married | 6060 (41.4) | 20.1 (19.1–21.1) | |
| **Age at first sex (in years)** | | | <0.001* |
| Never had sex | 4044 (27.6) | 14.2 (13.1–15.3) | |
| <18 | 2671 (18.3) | 41.9 (40.1–43.8) | |
| 18+ | 7919 (54.1) | 62.3 (61.2–63.4) | |
| **Health insurance coverage** | | | 0.001* |
| Not covered | 2455 (16.8) | 42.5 (40.5–44.5) | |
| Covered | 12179 (83.2) | 46.1 (45.2–47.0) | |
| **Years lived in place of residence** | | | <0.001* |
| 5+ | 10428 (71.3) | 40.9 (40.0–41.9) | |
| <5 | 4206 (28.7) | 56.8 (55.2–58.3) | |
| **Literacy** | | | 0.002* |
| Cannot read at all | 2176 (14.9) | 42.5 (40.4–44.6) | |
| Able to read | 12439 (85.1) | 46.0 (45.2–46.9) | |
| **Sex of household head** | | | <0.001* |
| Male | 10045 (68.6) | 49.0 (48.1–50.0) | |
| Female | 4589 (31.4) | 37.7 (36.3–39.1) | |
| **Knowledge of HIV test kits** | | | <0.001* |
| No | 11978 (81.9) | 43.1 (42.2–44.0) | |
| Yes | 2656 (18.2) | 56.4 (54.5–58.3) | |
| **Socioeconomic status** | | | <0.001* |
| Low | 4879 (33.3) | 47.4 (46.0–48.8) | |
| Medium | 4890 (33.4) | 46.4 (45.0–47.8) | |
| High | 4865 (33.2) | 42.6 (41.2–44.0) | |
| **Media access** | | | 0.003* |
| No | 2551 (17.4) | 42.8 (40.9–44.8) | |

*(Continued)*

**Table 1.** (Continued)

| Variable | Number of women (%) | Weighted prevalence of prenuptial HIV/AIDS testing; % (95%CI) | P value |
|---|---|---|---|
| Yes | 12083 (82.6) | 46.0 (45.1–46.9) | |
| **Religion** | | | 0.212 |
| Catholic | 5506 (37.6) | 44.8 (43.5–46.1) | |
| Other Christians | 8710 (59.5) | 46.0 (45.0–47.1) | |
| Muslims | 287 (2.0) | 45.8 (40.1–51.6) | |
| Others | 131 (0.9) | 38.6 (30.5–47.3) | |
| **Enlightenment** | | | 0.037* |
| Low | 7158 (48.9) | 44.8 (43.6–46.0) | |
| Medium | 3121 (21.3) | 44.7 (43.0–46.5) | |
| High | 4355 (29.8) | 47.1 (45.6–48.6) | |
| **Place of residence** | | | <0.001* |
| Urban | 3551 (24.3) | 48.5 (46.9–50.2) | |
| Rural | 11083 (75.7) | 44.5 (43.6–45.4) | |
| **Geographical region** | | | <0.001* |
| Kigali | 1921 (13.1) | 51.5 (49.2–53.7) | |
| South | 3482 (23.8) | 44.9 (43.2–46.5) | |
| West | 3312 (22.6) | 48.9 (47.2–50.6) | |
| North | 2294 (15.7) | 45.1 (43.0–47.1) | |
| East | 3625 (24.8) | 40.1 (38.4–41.7) | |
| **Community poverty** | | | <0.001* |
| Low | 7297 (49.9) | 48.3 (47.1–49.4) | |
| High | 7337 (50.1) | 42.7 (41.5–43.8) | |
| **Community media use** | | | <0.001* |
| Low | 7252 (49.6) | 43.6 (42.5–44.8) | |
| High | 7382 (50.4) | 47.3 (46.2–48.5) | |
| **Community illiteracy** | | | <0.001* |
| Low | 7297 (49.9) | 47.7 (46.5–48.8) | |
| High | 7337 (50.1) | 43.3 (42.1–44.4) | |
| **Community nativity** | | | <0.001* |
| Low | 7179 (49.1) | 49.8 (48.5–50.9) | |
| High | 7455 (50.9) | 41.4 (40.3–42.6) | |
| **Community knowledge of HIV test kits** | | | <0.001* |
| Low | 7269 (49.7) | 41.9 (40.8–43.1) | |
| High | 7365 (50.3) | 49.0 (47.8–50.1) | |

*Significant at p<0.05

prenuptial HIV/AIDS testing has been explained by the community-level factors. Furthermore, VPC for community-level factors was estimated to 4.9%. See Table 2 below for the details.

## Measures of associations (fixed effects)

Results from Table 3 shows Model IV was selected as the best model because it has the least AIC value. Women aged 25–34 years had 1.48 times higher odds (OR = 1.48; 95%CI: 1.30–1.69), while those >34 years had reduction in the odds (OR = 0.43; 95%CI: 0.37–0.50) of prenuptial HIV/AIDS testing respectively, in comparison to women aged <25 years. Women with primary and secondary+ education had 1.31 (95%CI: 1.09–1.59) and 1.56 (95%CI: 1.25–

**Table 2. Random effect estimates of contextual factors associated with prenuptial HIV/AIDS testing among reproductive-aged women in Rwanda.**

| Random-effect | Model I | Model II | Model III | Model IV |
|---|---|---|---|---|
| **Community-level** | | | | |
| Variance (95% CI) | 0.26 (0.21–0.31)* | 0.37 (0.30–0.45)* | 0.20 (0.16–0.25)* | 0.32 (0.26–0.40)* |
| VPC (%) | 5.8 | 5.7 | 4.4 | 4.9 |
| Explained variance (PCV) | Reference | -42.3% | 23.1% | -23.1% |
| MOR | 1.63 | 1.78 | 1.53 | 1.71 |
| ICC (%) | 7.4 (6.1–8.9) | 10.1 (8.4–12.1) | 5.7 (4.6–7.0) | 8.8 (7.2–10.7) |
| **Model fit statistics** | | | | |
| AIC | 19459.1 | 14971.5 | 19397.4 | 14948.1 |
| BIC | 19474.3 | 15168.4 | 19488.3 | 15220.8 |
| Log likelihood | -9727.6 | -7459.7 | -9686.7 | -7438.1 |
| **Sample size** | | | | |
| Community-level | 500 | 500 | 500 | 500 |
| Individual-level | 14386 | 14367 | 14386 | 14367 |

Model I–empty null model, baseline model without any explanatory variables (unconditional model)

Model II–adjusted for only individual-level factors

Model III–adjusted for only community-level factors

Model IV–adjusted for individual- and community-level factors (full model)

VPC Variance Partition Coefficient, AIC Akaike's Information Criterion, BIC Bayesian Information Criterion, PCV Proportional Change in Variance, ICC Intra-class correlation

*Significant at p<0.05

1.95) times higher odds of prenuptial HIV/AIDS testing respectively, in comparison to uneducated women. In addition, women who got married or had first sex at an adult age (18+ years), had higher odds of prenuptial HIV/AIDS testing, in comparison to women who got married before age 18 years or never had sex respectively. Women's individual and community nativity, was significantly associated with prenuptial HIV testing in Rwanda. Those who had knowledge of HIV test kits, had 1.45 times higher odds of prenuptial HIV/AIDS testing, in comparison to women with no knowledge of HIV test kits (OR = 1.45; 95%CI: 1.30–1.63). The enlightened women had increased odds of prenuptial HIV/AIDS testing, when compared with those with low enlightenment. The geographical region of respondents was significantly associated with prenuptial HIV/AIDS testing.

## Discussion

Prenuptial HIV testing among reproductive-aged Rwandese women was examined using the 2019–20 nationally representative dataset. The first step toward treatment and care of HIV victims is testing, which is a crucial part of HIV prevention strategies. It is necessary to be aware of one's HIV status in order to seek and receive medical care, including antiretroviral therapy. The prevalence of prenuptial HIV/AIDS testing was approximately 46%. This is higher than the coverage of HIV testing among adolescents and young women in Nigeria (25.4%) [39]. In a previous study conducted in SSA countries, the uptake of HIV testing among women was lowest in Chad (1%) and highest in Rwanda (76%), the average county-level HIV testing for women was approximately 29% [40]. This result shows that much is still required to achieve the recommended testing rate. In 2017, Rwandan Ministry of Health recommended HIV self-testing as an additional strategy key populations, such as women. With only one-thirds of unmarried women reporting abstinence from sex, it is possible that premarital sexual exposure

**Table 3. Fixed effect of individual compositional and community-level factors associated with prenuptial HIV/AIDS testing among reproductive-aged women in Rwanda.**

| Fixed-effect | Model I | Model II | Model III | Model IV |
|---|---|---|---|---|
| **Age (in years)** | | | | |
| <25 | | 1.00 | | 1.00 |
| 25–34 | | 1.50 (1.31–1.71)* | | 1.48 (1.30–1.69)* |
| >34 | | 0.43 (0.37–0.50)* | | 0.43 (0.37–0.50)* |
| **Education** | | | | |
| No formal education | | 1.00 | | 1.00 |
| Primary | | 1.30 (1.07–1.57)* | | 1.31 (1.09–1.59)* |
| Secondary+ | | 1.55 (1.24–1.94)* | | 1.56 (1.25–1.95)* |
| **Wealth quintiles** | | | | |
| Poorest | | 1.00 | | 1.00 |
| Poorer | | 1.07 (0.93–1.22) | | 1.06 (0.93–1.22) |
| Middle | | 1.15 (0.99–1.33) | | 1.14 (0.98–1.31) |
| Richer | | 1.11 (0.95–1.29) | | 1.08 (0.93–1.26) |
| Richest | | 0.95 (0.80–1.14) | | 0.92 (0.76–1.10) |
| **Working status** | | | | |
| Not employed | | 1.00 | | 1.00 |
| Employed | | 0.99 (0.90–1.09) | | 0.99 (0.90–1.10) |
| **Age at first marriage (in years)** | | | | |
| <18 | | 1.00 | | 1.00 |
| 18–24 | | 1.65 (1.38–1.96)* | | 1.64 (1.38–1.95)* |
| >24 | | 4.00 (3.25–4.92)* | | 3.95 (3.21–4.86)* |
| Never married | | 0.27 (0.22–0.33)* | | 0.27 (0.22–0.33)* |
| **Age at first sex (in years)** | | | | |
| Never had sex | | 1.00 | | 1.00 |
| <18 | | 2.47 (2.09–2.91)* | | 2.47 (2.09–2.91)* |
| 18+ | | 3.48 (2.97–4.07)* | | 3.48 (2.98–4.08)* |
| **Health insurance coverage** | | | | |
| Not covered | | 1.00 | | 1.00 |
| Covered | | 1.07 (0.96–1.21) | | 1.08 (0.96–1.21) |
| **Years lived in place of residence** | | | | |
| 5+ | | 1.00 | | 1.00 |
| <5 | | 1.33 (1.20–1.47)* | | 1.31 (1.18–1.45)* |
| **Literacy** | | | | |
| Cannot read at all | | 1.00 | | 1.00 |
| Able to read | | 1.06 (0.90–1.25) | | 1.06 (0.90–1.24) |
| **Sex of household head** | | | | |
| Male | | 1.00 | | 1.00 |
| Female | | 0.89 (0.81–0.98)* | | 0.88 (0.80–0.97)* |
| **Knowledge of HIV test kits** | | | | |
| No | | 1.00 | | 1.00 |
| Yes | | 1.47 (1.32–1.65)* | | 1.45 (1.30–1.63)* |
| **Socioeconomic status** | | | | |
| Low | | 1.00 | | 1.00 |
| Medium | | 1.02 (0.86–1.22) | | 1.11 (0.89–1.40) |
| High | | 0.88 (0.73–1.05) | | 0.99 (0.74–1.31) |
| **Media access** | | | | |

(*Continued*)

**Table 3.** (Continued)

| Fixed-effect | Model I | Model II | Model III | Model IV |
|---|---|---|---|---|
| No | | 1.00 | | 1.00 |
| Yes | | 1.11 (0.98–1.26) | | 1.13 (0.99–1.28) |
| **Enlightenment** | | | | |
| Low | | 1.00 | | 1.00 |
| Medium | | 1.17 (1.04–1.32)* | | 1.16 (1.03–1.31)* |
| High | | 1.55 (1.36–1.75)* | | 1.55 (1.37–1.75)* |
| **Place of residence** | | | | |
| Urban | | | 1.00 | 1.00 |
| Rural | | | 1.24 (1.06–1.46)* | 1.13 (0.90–1.43) |
| **Geographical region** | | | | |
| Kigali | | | 1.00 | 1.00 |
| South | | | 0.97 (0.79–1.19) | 0.98 (0.75–1.27) |
| West | | | 1.14 (0.93–1.40) | 1.31 (1.01–1.70)* |
| North | | | 0.99 (0.79–1.23) | 0.94 (0.71–1.24) |
| East | | | 0.74 (0.60–0.91)* | 0.74 (0.57–0.96)* |
| **Community poverty** | | | | |
| Low | | | 1.00 | 1.00 |
| High | | | 0.92 (0.80–1.06) | 0.86 (0.71–1.05) |
| **Community media use** | | | | |
| Low | | | 1.00 | 1.00 |
| High | | | 1.05 (0.93–1.19) | 0.99 (0.84–1.15) |
| **Community illiteracy** | | | | |
| Low | | | 1.00 | 1.00 |
| High | | | 0.98 (0.86–1.11) | 1.02 (0.86–1.21) |
| **Community nativity** | | | | |
| Low | | | 1.00 | 1.00 |
| High | | | 0.72 (0.64–0.82)* | 0.86 (0.73–0.99)* |
| **Community knowledge of HIV test kits** | | | | |
| Low | | | 1.00 | 1.00 |
| High | | | 1.18 (1.04–1.34)* | 1.13 (0.97–1.33) |

Model I–empty null model, baseline model without any explanatory variables (unconditional model)

Model II–adjusted for only individual-level factors

Model III–adjusted for only community-level factors

Model IV–adjusted for individual- and community-level factors (full model)

*Significant at p<0.05

may be responsible for low uptake of prenuptial HIV testing, as some women may feel that HIV screening before marriage is needless as they may already have been sexually exposed especially with the partner.

The results showed higher odds of prenuptial HIV/AIDS testing among educated, enlightened and women having knowledge of HIV test kits. In a previous study, participants with higher educational attainment, exposed to media and those who had HIV knowledge, reported higher HIV testing [13]. In other studies conducted in Ethiopia and Nigeria, access to media and knowledge of HIV testing place were found as positive factors associated with premarital HIV testing among women [19, 39]. Women who have completed secondary school in 29 SSA countries were more likely to have ever undergone an HIV test than those who have only

completed primary school or have no formal education [40]. In a study conducted in Uganda, respondents who had good knowledge on HIV/AIDS, had higher odds utilize HIV testing services [41]. It is believed that educated and enlightened women will have better understanding of the need for prenuptial HIV testing, especially that the knowledge of HIV status will be key in timely enrolment for treatment. The fear of losing the potential spouse, in the event of HIV positive results and stigmatization may also scare people from HIV testing. This study found reduction in the odds of prenuptial HIV/AIDS testing among women from female headed households.

In a previous study, youth from communities with a high proportion of educated people had increased likelihood of HIV testing and counseling than their counterparts from communities with low percentage of educate people [42]. It is possible that young people from communities with a high proportion of educated youth will learn from others about the value of using HIV testing and counseling services as well as how to access them. Additionally, educated youth are more likely to comprehend health messages and request services. This study found significant community effects in addition to significant individual-level factors associated with prenuptial HIV testing. Women who had lived less than 5 years in the place of residence and those from high community nativity had reduction in the odds of prenuptial HIV/AIDS testing. The reason for differences in prenuptial HIV testing by nativity is unclear. However, a qualitative study may be used to decipher the circumstances surrounding the inequalities in prenuptial HIV testing between native and non-native women respectively.

Furthermore, region was linked with prenuptial HIV/AIDS testing. This is similar with the results from a previous study which identified a spatial variation of HIV counseling and testing in Ethiopia [43]. Another study conducted in Nigeria reported regional variation in HIV testing and counseling among young respondents [44]. The spatial variation of prenuptial HIV/AIDS testing could be explained on the basis of spatial clustering of HIV awareness campaigns and presence of interventions. The detection of regional differences in prenuptial HIV/AIDS testing could mean that HIV knowledge and education may be inadequate in some geographical region. This may also be connected to the fact that, given that the majority of HIV intervention programs are donor-driven, the success of implementing various HIV-related programs, such as the expansion of HIV testing and counseling, greatly depends on the ability of local implementers to use resources efficiently and implement interventions in accordance with local contexts. Additionally, these variations might be accounted for by the significant regional variations in enlightenment, education, exposure to media and HIV/AIDS awareness campaign. It is critical to prioritize and develop targeted prevention programs based on the geographic variation in HIV testing and counseling uptake in order to improve its utilization and lower HIV infection rates in high-risk areas.

Respondents with history of sexual exposure had greater odds of prenuptial HIV/AIDS testing, irrespective of their age at first sex. This is consistent with the report from a previous study which showed that the odds of ever having an HIV test were significantly higher for those who ever had sex [13]. In a previous study, never-married women reported lower HIV testing rates in the majority of SSA countries. In that study, the never-married respondents included those who reported having never had sex and are also the youngest, so HIV testing uptake was expected to be lower [40]. Higher prevalence of sexually active respondents reported HIV testing than women who have never had sex [40]. This study found women who married at adult age (18+ years) had higher odds of prenuptial HIV/AIDS testing, when compared with those who married before the adult age (<18 years). Furthermore, the age of a woman was significantly associated with prenuptial HIV/AIDS testing. An earlier study in SSA countries found that the age distribution of women who receive HIV testing tended to take the shape of an inverse U, as HIV testing uptake peaked between the ages of 20 and 34, and was

lower at either end of the age spectrum, especially among teenagers [40]. Similar pattern of age distribution in prenuptial HIV testing was found in this study. The younger women may lack the empowerment and decision-making power to request for HIV testing before marriage.

## Strengths and limitations

Large, nationally representative dataset with high response rate were used in this study. The findings can be applied to women aged 15 to 49 year. As a result of the hierarchical nature of DHS data, this study conducted multilevel modeling. In spite of these strengths, the study variables were self-reported, making them prone to social desirability biases. Furthermore, primary sampling unit (PSU) was used to define communities. Furthermore, variables on the availability and proximity of health infrastructures were not captured by DHS and could be critical in HIV testing. Only association and not causality can be established in this study due to the cross-sectional nature of the data.

## Conclusion

There was low uptake of prenuptial HIV testing among Rwandese women. In addition, individual compositional and community-level factors were associated with prenuptial HIV testing. To achieve the UNAIDS first '95' target, HIV testing should be considerably scaled-up. Several interventions and policies could be considered to increase HIV testing uptake. Additionally, mobile HIV testing campaigns may be especially effective in increasing HIV testing coverage. The Rwandan Ministry of Health and other stakeholders should consider prenuptial HIV testing factors when raising HIV testing awareness among women. To improve access to HIV testing services, we recommend a combination of strategies. The interventions should aim to increase young people's HIV knowledge, reduce misconceptions and stigmatization and increase access to HIV testing through home-based testing and an "opt-out" strategy at the point of care.

## Acknowledgments

The authors appreciate the Demographic and Health Survey for the approval and access to the original data.

## Author Contributions

**Conceptualization:** Michael Ekholuenetale, Olah Uloko Owobi, Amadou Barrow.

**Data curation:** Michael Ekholuenetale, Olah Uloko Owobi, Amadou Barrow.

**Formal analysis:** Michael Ekholuenetale, Olah Uloko Owobi, Amadou Barrow.

**Investigation:** Michael Ekholuenetale, Amadou Barrow.

**Methodology:** Michael Ekholuenetale, Olah Uloko Owobi, Amadou Barrow.

**Project administration:** Michael Ekholuenetale.

**Resources:** Michael Ekholuenetale.

**Software:** Michael Ekholuenetale.

**Supervision:** Michael Ekholuenetale, Amadou Barrow.

**Validation:** Michael Ekholuenetale, Olah Uloko Owobi, Amadou Barrow.

**Visualization:** Michael Ekholuenetale, Olah Uloko Owobi, Amadou Barrow.

**Writing – original draft:** Michael Ekholuenetale, Olah Uloko Owobi, Amadou Barrow.

**Writing – review & editing:** Michael Ekholuenetale, Olah Uloko Owobi, Amadou Barrow.

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
