## [Decision Letter · Decision Letter 0]

17 Oct 2022

PGPH-D-22-01238

Achieving the UNAIDS first ‘95’ in prenuptial HIV/AIDS testing among reproductive-aged Rwandese women: a multilevel analysis of 2019-20 population-based data

Dear Dr. Barrow,

Thank you for submitting your manuscript to PLOS Global Public Health. After careful consideration, we feel that it has merit but does not fully meet PLOS Global Public Health’s publication criteria as it currently stands. Therefore, we invite you to submit a revised version of the manuscript that addresses the points raised during the review process.

We look forward to receiving your revised manuscript.

Kind regards,

Olatunji O Adetokunboh, MD, PhD

Academic Editor

Journal Requirements:

1. Please send a completed 'Competing Interests' statement, including any COIs declared by your co-authors. If you have no competing interests to declare, please state "The authors have declared that no competing interests exist". Otherwise please declare all competing interests beginning with the statement "I have read the journal's policy and the authors of this manuscript have the following competing interests:" Achieving the UNAIDS first ‘95’ in prenuptial HIV/AIDS testing among reproductive-aged Rwandese women: a multilevel analysis of 2019-20 population-based data
 2. Tables should not be uploaded as individual files. Please remove these files and include the Tables in your manuscript file as editable, cell-based objects. For more information about how to format tables, see our guidelines: https://journals.plos.org/globalpublichealth/s/tables

Additional Editor Comments (if provided):

Reviewers' comments:

Reviewer's Responses to Questions

**Comments to the Author**

1. Does this manuscript meet PLOS Global Public Health’s publication criteria? Is the manuscript technically sound, and do the data support the conclusions? The manuscript must describe methodologically and ethically rigorous research with conclusions that are appropriately drawn based on the data presented.

Reviewer #1: Partly

Reviewer #2: No

2. Has the statistical analysis been performed appropriately and rigorously?

Reviewer #1: No

Reviewer #2: I don't know

3. Have the authors made all data underlying the findings in their manuscript fully available (please refer to the Data Availability Statement at the start of the manuscript PDF file)?

Reviewer #1: Yes

Reviewer #2: Yes

4. Is the manuscript presented in an intelligible fashion and written in standard English?

Reviewer #1: Yes

Reviewer #2: Yes

5. Review Comments to the Author

Reviewer #1: I congratulate the authors for selecting a topic that has strong policy implications for achieving the UNAIDS' first ‘95’ target. The manuscript does not have page and line numbers, for which I will only focus on major comments instead of specific ones. My major comments are regarding the manuscript's methodological aspects.

First, the prevalence of prenuptial HIV/AIDS testing was found to be high, i.e., 45.9% (95% CI: 44.8-47.1). In the situation where the binary outcome is common, i.e., the prevalence is greater than 10%, the odds ratio (OR) estimated from binary logistic regression overestimates or jeopardizes the true measure of association (for more information, please see Ranganathan et al. 2015 and Tamhane et al. 2016). In such cases, a prevalence ratio (PR) instead of OR is the appropriate option. Authors should consider using Poisson regression models with robust variance, reporting Adjusted Prevalence Ratios (APR) and 95% Confidence Intervals (CIs). The authors may also consider using log-binomial or other appropriate methods for calculating the adjusted PR.

Second, the authors correctly stated that given the hierarchical nature of the DHS data, a multilevel model would be an appropriate analytical approach. Multilevel analysis is also a suitable approach to take into account the social contexts as well as the individual respondents. My major concern is about the authors’ choice of levels for the analyses. To me, it was not clear how the levels were selected and this needs to be described and supported. Have the authors used any theoretical framework or solely relied on the DHS sampling frame used for Rwanda? For example, if the authors had used a conceptual framework of health services utilization, viz., Aday (1974), Andersen and Newman (1973), Andersen (1995), then following levels would seem suitable:

1. Individuals/household (n=13,000)

2. Clusters/EAs (n=500)

3. Districts (n=30)

4. Provinces (n=5)

Lastly, it is not clear how the authors developed the socioeconomic status (SES) index. Have they used rural location and unemployment in developing the SES index? How it differs from the standard asset score needs to be described. Including "unemployment" in the standard list for developing asset score may be problematic, as the assets indicate the household's long-term socioeconomic status, and unemployment may indicate an individual-level recent phenomenon. Also, keeping location (rural/urban) a separate indicator would give you more information on the effect of place of residence on HIV testing services use.

References:

Aday LA, Andersen RM. A framework for the study of access to medical care. Health Serv Res 1974;9(3):208-220.

Andersen RM, Newman JF. Societal and individual determinants of medical care utilization in the United States. Milbank Memorial Fund Quarterly– Health and Society 1973;51(1):95-124.

Andersen RM. 1995. Revisiting the Behavioral Model and access to medical care: does it matter? Journal of Health and Social Behavior 36: 1–10.

Ranganathan P, Aggarwal R, Pramesh CS. Common pitfalls in statistical analysis: Odds versus risk. Perspect Clin Res. 2015;6(4):222. DOI: 10.4103/2229-3485.167092.

Tamhane AR, Westfall AO, Burkholder GA, Cutter GR. Prevalence odds ratio versus prevalence ratio: choice comes with consequences. Stat Med. 2016;35(30):5730–5. DOI: 10.1002/sim.7059.

Reviewer #2: Comments are uploaded in the file attached. I have the following comments:

1. The article has no line numbers

2. Abstract, methods: include the location of the data if one is interested in re-analyzing the data

3. Abstract, results: include the odds ratio, 95% confidence intervals, and p-values of all the factors found

4. Abstract, conclusion: Were the recommendations derived from this study? What was their odds ratio and p-values to show that they worked?

5. Main results: provide p-values for the variables in the models, besides having them in the table

6. References: Some of them are duplicated eg 4 and 6, 23 and 25.

7. Some of the tables are more than a page long, try to shorten them.

Thank you,

6. PLOS authors have the option to publish the peer review history of their article (what does this mean?). If published, this will include your full peer review and any attached files.

**Do you want your identity to be public for this peer review?** For information about this choice, including consent withdrawal, please see our Privacy Policy.

Reviewer #1: No

Reviewer #2: No

---

## [Decision Letter · Decision Letter 1]

5 Dec 2022

PGPH-D-22-01238R1

Achieving the UNAIDS first ‘95’ in prenuptial HIV/AIDS testing among reproductive-aged Rwandese women: a multilevel analysis of 2019-20 population-based data

Dear Dr. Barrow,

Thank you for submitting your manuscript to PLOS Global Public Health. After careful consideration, we feel that it has merit but does not fully meet PLOS Global Public Health’s publication criteria as it currently stands. Therefore, we invite you to submit a revised version of the manuscript that addresses the points raised during the review process.

We look forward to receiving your revised manuscript.

Kind regards,

Olatunji O Adetokunboh, MD, PhD

Academic Editor

Journal Requirements:

Additional Editor Comments (if provided):

Reviewers' comments:

Reviewer's Responses to Questions

**Comments to the Author**

1. If the authors have adequately addressed your comments raised in a previous round of review and you feel that this manuscript is now acceptable for publication, you may indicate that here to bypass the “Comments to the Author” section, enter your conflict of interest statement in the “Confidential to Editor” section, and submit your "Accept" recommendation.

Reviewer #1: (No Response)

Reviewer #2: All comments have been addressed

2. Does this manuscript meet PLOS Global Public Health’s publication criteria? Is the manuscript technically sound, and do the data support the conclusions? The manuscript must describe methodologically and ethically rigorous research with conclusions that are appropriately drawn based on the data presented.

Reviewer #1: Partly

Reviewer #2: Yes

3. Has the statistical analysis been performed appropriately and rigorously?

Reviewer #1: No

Reviewer #2: Yes

4. Have the authors made all data underlying the findings in their manuscript fully available (please refer to the Data Availability Statement at the start of the manuscript PDF file)?

Reviewer #1: Yes

Reviewer #2: Yes

5. Is the manuscript presented in an intelligible fashion and written in standard English?

Reviewer #1: Yes

Reviewer #2: Yes

6. Review Comments to the Author

Reviewer #1: I thank the authors for incorporating feedback and detailed responses. I am highlighting below the issues that require the authors’ attention:

First, the authors’ statements that Poisson regression is for count data and not suitable for modeling our binary outcome are not correct. Poisson models can be used for the binary outcome and, in fact, there are options for Poisson models (viz., using robust error variances or quasi-Poisson models) that provide correct estimates and are a better alternative for the analysis of cross-sectional studies with binary outcomes than logistic regression. Given the high prevalence of prenuptial HIV/AIDS testing, using the odds ratio instead of the prevalence ratio (or interpreting the odds ratio as if it were a prevalence ratio) is not a technically sound approach not only in terms of the possible overestimation, but also because confounding may not be appropriately controlled. There are published articles on how logit models are widely misused in public health/epidemiological studies. Apart from the references I already stated in my original feedback, the authors may review the highly cited article by Barros & Hirakata (2003) from http://www.biomedcentral.com/1471-2288/3/21 in this regard.

Second, you are getting an error message while using log-binomial models in multilevel analysis because, while using meglm, a log link is not allowed with binomial or Bernoulli family due to convergence issues. The authors may review the help file for meglm where the allowable combinations for links and families are provided in a table.

Third, the authors responded that their approach to developing the SES index has also been used in previous studies. Given the unusual approach used for the SES index, I’d recommend the authors to provide a few references to support this claim. In the manuscript, the authors cited references # 29 and 30. These studies used PCA for developing household wealth indices, but neither used rurality, lack of formal education, unemployment, and poorest quintile (which, in turn, was derived using PCA itself) to create an index using PCA. Also, information on respondent’s education and employment multiple times under different variables (viz., educational attainment, working status, SES, Enlightenment). Have the authors tested for multicollinearity in their regression model?

Finally, the manuscript would be benefitted from proofreading and copyediting as there are issues with punctuation and subject-verb agreement throughout the manuscript.

Reviewer #2: I have no further comments

7. PLOS authors have the option to publish the peer review history of their article (what does this mean?). If published, this will include your full peer review and any attached files.

**Do you want your identity to be public for this peer review?** For information about this choice, including consent withdrawal, please see our Privacy Policy.

Reviewer #1: No

Reviewer #2: **Yes: **Dr. Silvia Awor

---

## [Editor Report · Decision Letter 2]

17 Jan 2023

Achieving the UNAIDS first ‘95’ in prenuptial HIV/AIDS testing among reproductive-aged Rwandese women: a multilevel analysis of 2019-20 population-based data

PGPH-D-22-01238R2

Dear Mr. Barrow,

We are pleased to inform you that your manuscript 'Achieving the UNAIDS first ‘95’ in prenuptial HIV/AIDS testing among reproductive-aged Rwandese women: a multilevel analysis of 2019-20 population-based data' has been provisionally accepted for publication in PLOS Global Public Health.

Best regards,

Olatunji O Adetokunboh, MD, PhD

Academic Editor